# Deep Learning with Attention Mechanisms for Road Weather Detection

**DOI:** 10.3390/s23020798

**Published:** 2023-01-10

**Authors:** Madiha Samo, Jimiama Mosima Mafeni Mase, Grazziela Figueredo

**Affiliations:** School of Computer Science, University of Nottingham, Nottingham NG7 2RD, UK

**Keywords:** computer vision, deep learning, image classification, loss functions, vision transformers, weather detection, autonomous vehicles

## Abstract

There is great interest in automatically detecting road weather and understanding its impacts on the overall safety of the transport network. This can, for example, support road condition-based maintenance or even serve as detection systems that assist safe driving during adverse climate conditions. In computer vision, previous work has demonstrated the effectiveness of deep learning in predicting weather conditions from outdoor images. However, training deep learning models to accurately predict weather conditions using real-world road-facing images is difficult due to: (1) the simultaneous occurrence of multiple weather conditions; (2) imbalanced occurrence of weather conditions throughout the year; and (3) road idiosyncrasies, such as road layouts, illumination, and road objects, etc. In this paper, we explore the use of a focal loss function to force the learning process to focus on weather instances that are hard to learn with the objective of helping address data imbalances. In addition, we explore the attention mechanism for pixel-based dynamic weight adjustment to handle road idiosyncrasies using state-of-the-art vision transformer models. Experiments with a novel multi-label road weather dataset show that focal loss significantly increases the accuracy of computer vision approaches for imbalanced weather conditions. Furthermore, vision transformers outperform current state-of-the-art convolutional neural networks in predicting weather conditions with a validation accuracy of 92% and an F1-score of 81.22%, which is impressive considering the imbalanced nature of the dataset.

## 1. Introduction

Different types of weather severely affect traffic flow, driving performance, and vehicle and road safety [1]. Statistics from the Federal Highway Administration show that increased amount of accidents and congestion are usually directly associated with hostile weather [2]. As a result, there is the need for advanced intelligent systems that can accurately and automatically detect weather conditions to support safe driving, traffic safety risk assessment, autonomous vehicles, and effective management of the transport network. Deep learning has emerged as one of the main approaches used for automatic weather recognition as evidenced by the related work in Section 2. The state-of-the-art literature mostly employs convolutional neural networks (CNN), which are trained on outdoor weather images and subsequently label new images with a single weather class. This type of classification for roads, however, produces less accurate results, as multiple weather types are likely to occur simultaneously. For example, Figure 1 shows multiple weather conditions (i.e., sunny and wet) present in a single scenario. Another limitation found in the current related work is that deep learning models are mostly trained on balanced and high variance weather datasets. This oversimplifies road weather conditions, which are characterised by highly imbalanced and more complex scenarios, such as road layouts, interacting elements, vehicles, people, and different illumination conditions. The representation learning, therefore, becomes compromised as road elements that could potentially allow for a more specific type of learning for the road problem are not included. There is also currently no research study investigating intelligent strategies for multi-label, highly imbalanced and complex road scenarios, such as dynamic pixel-based weighting. This drives the motivation of this study to propose a publicly available realistic multi-label road weather dataset and employ vision transformers based on focal loss to address class imbalance and road idiosyncrasies.

The main contributions of this study are:A multi-label transport-related dataset consisting of seven weather conditions: sunny, cloudy, foggy, rainy, wet, clear, and snowy to be used for road weather detection research.Assessment of different state-of-the-art computer vision models in addressing multi-label road weather detection, using our dataset as a benchmark.Evaluation of the effectiveness of focal loss function to increase model accuracy for unbalanced classes and hard instances.Implementing transformer vision models to assess the efficiency of their attention mechanism (assigning dynamic weights to pixels) in addressing road weather idiosyncrasies.

This paper is organised as follows. In Section 2, we review the literature on weather detection using deep learning techniques and describe the focal loss function to handle imbalanced data and difficult-to-classify instances. Subsequently, we provide an overview of the CNN architectures explored in this paper. Section 3 describes vision transformers in comparison to CNN networks. Section 4 introduces our novel multi-label road weather dataset, describes the vision transformer models implemented in this paper, and presents the design of our experiments and evaluation protocols. In Section 5, the results are presented along with a discussion, and Section 6 concludes the paper and establishes the opportunity for future work.

## 2. Background

### 2.1. Related Work

The rapid evolution and widespread application of sensors (e.g., onboard cameras) has led to large volumes of data streams constantly being generated in transportation. Deep learning approaches have emerged as suitable approaches to address big data problems as they reduce the dependency on human experts and learn high-level features from data in an incremental manner. Specifically, for weather recognition tasks, convolution neural networks have been vastly explored by many researchers.

Kang et al. [3] introduced a weather classification framework based on GoogleNet to recognise four weather conditions—hazy, snowy, rainy and others. Their framework was trained using the general MWI weather dataset [4] and achieved 92% accuracy. The model outperformed multiple kernel learning-based approaches [4] and AlexNet CNN [5]. Similarly, An et al. [6] explored ResNet and Alexnet coupled with support vector machines for weather classification. The authors evaluated the models using several multi-class weather datasets. The ResNet architecture outperformed AlexNet with a classification accuracy of 92% and 88% for sunny and cloudy classes, respectively.

In Khan et al. [7], the authors developed deep learning models to recognise both weather and surface conditions based on images from roadside webcams. Their dataset consists of three weather conditions (clear, light snow and heavy snow) and three road surface conditions (dry, snowy and wet). They explored different CNN architectures, including ResNet-18, GoogleNet and AlexNet, and amongst the architectures, ResNet-18 achieved the best detection accuracy with 97% for weather and 99% for road surface conditions.

Guerra et al. [8] introduced another multi-class Weather dataset called RFS, consisting of three classes—rainy, foggy and snowy. The authors also employed ResNet architecture to achieve 80.7% accuracy on their dataset. Later, Jabeen et al. [9] utilised inception CNN architecture for weather detection using a new multi-class weather dataset consisting of 2000 images belonging to three classes, namely foggy, rainy and clear. Their model achieved an average of 98% accuracy for the three classes.

Zhao et al. [10] employed CNNs coupled with recurrent networks on a multi-label weather dataset to address the problem of more than one weather condition existing in a single image. The dataset consists of five classes, including sunny, snowy, cloudy, rainy, and foggy. Their architecture achieved an average F-score of 87% for the five classes. However, the dataset used is a generalised weather dataset that is not specific to roads.

Recently, Xia et al. [11] explored ResNet CNNs to classify images in a multi-class weather dataset called WeatherDataset-4 into different weather conditions. WeatherDataset-4 dataset is made up of four major classes, including foggy, snowy, rainy and sunny. The authors achieved an average classification accuracy of 96.03%. In addition, Togacar et al. [12] employed GoogleNet and VGG16 Spiking Neural Networks (SNNs) for weather recognition. The weather dataset used by the authors consists of four classes: cloudy, rainy, sunny and sunrise. The features from GoogleNet and VGG16 are combined and trained using SNNs. The average classification result obtained with the combined CNNs and SNNs was 97.88%, which is much better than using the CNN models without SNNs.

In addition to the CNN models discussed above, due to significant improvements and success of transformers in the natural language processing tasks [13], researchers applied transformers to computer vision, with CNN being the fundamental component. In [14], Chen et al. proposed an image generative pretrained transformer (iGPT) method combined with self-supervised learning methods. This approach consists of pre-training followed by fine-tuning stage and was inspired by unsupervised representation learning for natural language. A sequence transformer is trained to auto-regressively predict pixels, without considering the 2D input structure. Therefore, in this approach, sequence transformer architecture is applied to predict pixels instead of language tokens. The proposed model achieved 96.2% accuracy, with a linear probe on CIFAR-10. The model achieves results comparable to CNN models in the image classification tasks. Similarly, Dosovitskiy et al. [15] proposed another model based on a transformer for image classification tasks named ViT. The ViT model applies a pure transformer directly to the image patch sequences in order to classify a complete image. Dosovitskiy et al. proposed various variants of ViT as ViT-Base, ViT-Large and ViT-Huge, where the base and large models are adopted directly from BERT [16]. Each variant uses brief notation to indicate the input patch size and the model size. For instance, ViT-L/32 means the model variant “Large” with an input patch size of 32 × 32. The largest model variant (ViT-H/14) achieved the highest accuracy of 99.5% for the CIFAR-10 dataset. The detailed architecture of ViT is discussed later in detail in Section 3. The models iGPT and ViT are both attempts to apply transformers to computer vision applications. However, there are quite a few differences between the two, as follows: (1) ViT only has a transformer encoder, while iGPT architecture is an encoder–decoder framework; (2) ViT uniformly divides the images into a sequence of patches, whereas iGPT takes a sequence of color palettes by clustering pixels as an input; (3) ViT is trained using a supervised image classification task, while iGPT uses auto-regressive self-supervised loss for training.

The classification performance achieved in the above studies for weather recognition is acceptable. However, the majority of the studies focused on multi-class classification, which could be unrepresentative of real-world weather conditions where more than one weather condition can occur simultaneously (as shown in the sample image in Figure 1). The few studies that employ multi-label classification [10] are either implemented on a general weather dataset or fail to make their datasets available for comparison and advancement. In addition, the studies use carefully selected outdoor images, which create well-balanced weather datasets. This oversimplifies the road weather detection problem, which is usually imbalanced in nature, e.g., icy and snowy weather conditions rarely occur in the United Kingdom (UK). The outdoor datasets also fail to include different lighting conditions and road characteristics, making them ungeneralizable to road weather images.

We address the above limitations by proposing a multi-label weather dataset for roads to address the problem of multiple weather existing in a single frame. In addition, as the weather data are inherently unbalanced, an attention mechanism needs to be provided to address those categories that are harder to learn, as those are more likely to be extreme (rare) conditions, and their misclassification by the intelligent systems should be minimised. Hence, the systematic approach followed in this study allows the model to focus more on less represented classes instead of data-dominated labels to prevent training a bias network. We also focus on feeding the model information about hard instances to avoid the gradient being outclassed by the accumulation of the losses of easy instances. Lastly, we focus on dynamically assigning weights to the pixels allowing the model to focus more on relevant features during classification, which can potentially increase the model’s efficiency for highly complex data. Specifically, the study involves identifying the potential of adapting weighted loss and focal loss function to deal with class imbalance problems and hard-to-learn instances in the dataset. The study also involves exploring vision transformer models allowing the model to focus more on relevant pixels only. To the best of our knowledge, this study is the first attempt to recognise the potential of weighted loss, focal loss and pixel-based attention mechanism for multi-label road weather classification.

### 2.2. Loss Functions Explored in This Study to Deal with Data Predicaments

Class Weighted Loss Function: The traditional cross entropy loss does not take into account the imbalanced nature of the dataset. The inherent assumption that the data are balanced often lead to fallacious results. Since the learning becomes biased towards majority classes, the model fails to learn meaningful features to identify the minority classes. Therefore, to overcome these issues, loss function can be optimised by assigning weights such that more attention is given to minority classes during training. Weights are assigned to each class such that the smaller the number of instances in a class, the greater the weight assigned to that class. For each class, Weight assigned to the class = Total images in dataset/Total images in that class. The weighted cross-entropy loss function is given by:
(1)$$\begin{array}{cc} L=\sum_{i=1}^{N}\sum_{c=1}^{C} & \omega_{c}\left[-\left(y_{c}*\left(p_{c}\right)+\left(1-y_{c}\right)*\left(1-p_{c}\right)\right)\right] \end{array}$$
where L is the total loss, c represents the class, i represents the training instance, while C and N represent total number of classes and instances, respectively. The y_c_ indicates the ground truth label for the class c, and p_c_ is the predicted probability that the given image belongs to class c, while ω_c_ represents the weight of the class c.Focal Loss Function: A focal loss function is a dynamically scaled cross-entropy loss function. Focal loss forces the model to focus on the hard misclassified examples during the training process [17]. For any given instance, the scaling factor of the focal loss function decays to zero as the loss decreases, thus allowing the model to rapidly focus on hard examples instead of assigning similar weights to all the instances. Focal loss function is given by
(2)$$\begin{array}{c} \mathrm{FL}\left(p_{o}\right)=-\alpha_{o}{\left(1-p_{o}\right)}^{\gamma }\log\left(p_{o}\right) \end{array}$$
where α and γ are hyperparameters such that setting γ greater than zero reduces relative loss for examples that are easily classified. The hyperparameter γ>= 0 and its value controls the loss for easy and hard instances, while α lies between [0,1] and addresses the class imbalance problem.

### 2.3. Deep Learning Architectures Investigated

Several state-of-the-art CNN architectures have been successfully proposed for image classification. Table 1 briefly describes the structure of state-of-the-art CNN architectures used in this study, including VGG19, GoogleNet, ResNet-152, Inception-v3, and EfficientNet-B7.

## 3. Vision Transformers

Transformers were initially introduced for Natural Language Processing (NLP) tasks [23], while image processing tasks usually relied on convolution neural networks. Recently, transformers have been adopted for computer vision tasks [15] and they are called vision transformers. Vision transformers are similar to NLP transformers, where patches of images are used instead of sentences. Images are broken down into a series of patches and transformed into embeddings, which can be easily fed into NLP transformers, similar to embeddings of words.

Conventional CNNs typically assign similar attention (weights) to all the pixels of an image during classification. As already proven in the field of NLP, introducing attention mechanisms such that higher weights are assigned to pixels of relevant information could lead to potentially better results and efficient models. Therefore, Vision Transformers (ViT) capture relationships between different parts of an image allowing the model to focus more on relevant pixels in classification problems. ViT computes relationships among pixels in small sections of the image (also known as patches) to reduce computation time instead of computing the relationship between each individual pixel. Each image is considered a sequence of patches of pixels. However, to retain the positional information, positional embeddings are added to the patch embeddings, as shown in Figure 2. These positional embeddings are important to represent the position of features in a flattened sequence; otherwise, the transformer will lose information about the sequential relationships between the patches. A positional embedding (PE) matrix is used to define the relative distance of all possible pairs in the given sequence of patch embeddings and is given by the formula:(3)$$\begin{array}{c} PE_{\left(pos,2_{i}\right)}=sin\left(pos/1000^{\left(2_{i}/d_{model}\right)}\right) \end{array}$$
$$\begin{array}{c} PE_{\left(pos,2_{i}+1\right)}=cos\left(pos/1000^{\left(2_{i}/d_{model}\right)}\right) \end{array}$$
where pos is the position of the feature in the input sequence, *i* is used to map column indices such that 0 <= i <=
d/2, and *d* is the dimension of the embedding space.

The results with the position embeddings are then fed to a transformer encoder for classification, as shown in Figure 3. The transformer encoder module consists of a Multi-Head Self Attention (MSA) layer and a Multi-Layer Perceptron (MLP) layer. The MSA layer splits the given input into multiple heads such that each head learns different levels of self-attention. The outputs are then further concatenated and passed through the MLP layer. The concatenated outputs from the MSA layer are normalised in the Norm layer and sent to the MLP layer for classification. The MLP layer consists of Gaussian Error Linear Unit (GELU) activation functions.

Figure 2 shows an overview of ViT. This section concludes by explaining in more detail the attention mechanism adopted by the MSA layer.

A typical attention mechanism is based on trainable vector pairs consisting of keys and values. A set of *k* key vectors is packed in a matrix *K* $\left(K\in R^{kxd}\right)$ such that the query vector $\left(q\in R^{d}\right)$ is matched against this set of *k* key vectors. The matching is based on inner dot products, which are then scaled and normalised. A softmax function is then applied to obtain *k* weights. The weighted sum of *k* value vectors then serve as an output of the attention. For self-attention, the vectors (Query, Key and Value) are calculated from a given set of *N* input vectors (i.e., patches of images) such that:

$Query=XW_{q},Key=XW_{k},Value=XW_{v}$, where $W_{q},W_{k},$ and $W_{v}$ are the linear transformations with the constraint k=N, indicating that the attention is computed between the given *N* input vectors.

The MSA layer refers to the “h” number of self-attention functions applied to the input, as follows: $Multihead\left(Q,K,V\right)=\left[head1,\ldots ,headh\right]W_{0}$, where *W* refers to the learnable parameter matrices. MSA computation is made such that query, key and value vectors are split into *N* vectors before applying self-attention. The self-attention process is then applied to each split vector individually. The independent attention modules are concatenated and linearly transformed.

We conclude this section by summarising the image classification process of ViT using the self-attention mechanism and encoder layer described above. Input images are split into patches of fixed sizes and multiplied with embedding matrices. Each patch is assigned a trainable positional embedding vector to remember the order of the input sequence before feeding the input to the transformer. The transformer uses constant vector size in all the layers so all the patches are flattened to map these dimensions using a trainable linear projection. Each encoder comprises two sub-layers. The first sub-layer allows the input to pass through the self-attention module while the outputs of the self-attention operation are then passed to a feed-forward neural network in the second sub-layer with output neurons for classifying the images. Skip connections and layer normalisation are also incorporated into the architecture for each sublayer of the encoder.

## 4. Experiments

### 4.1. Proposed Dataset Description

Due to a lack of a publicly available multi-label road weather dataset, we have created an open-source dataset consisting of road images depicting eight classes of weather and road surface conditions, i.e., sunny, cloudy, foggy, rainy, wet, clear, snowy and icy. The images are extracted from available online videos on YouTube captured and uploaded by ‘Alan Z1000sx’ (the YouTube account that owns the road-facing videos) using a video camera mounted on the dashboard of a heavy goods vehicle completing journeys across the UK (a sample video is available at [24]). The video clips captured different roads in the UK (i.e., motorways, urban roads, rural roads, and undivided highways), different weather conditions (i.e., sunny, cloudy, foggy, rainy, wet, clear, snowy and icy) and different lighting conditions (i.e., sunset, sunrise, morning, afternoon, night, and evening). We downloaded 25 videos uploaded by ‘Alan Z1000sx’ with an average duration of 8 min. We developed a python script to extract images from the videos every 10 s. A total of 2,498 images were extracted.

To annotate the images, we utilised an online annotation platform called Zooniverse [25]. In Zooniverse, volunteers assist researchers in data annotation and pattern recognition tasks. We created a project in Zooniverse for annotating the images, uploaded the images, specified the labels, and added volunteers to our project. Zooniverse provides an easy-to-use interface for annotating the images, as shown in Figure 4. As shown in the figure, each image could be assigned to more than one weather condition. The annotations were carried out by two volunteers. After annotating the images, Zooniverse offers an option to export the annotations to a comma-separated values file. Table 2 shows the distribution of the images in different weather conditions. The dataset is imbalanced with the majority of the images classified as clear and sunny, while icy is the least classified as UK roads are rarely icy. Six sample images from the dataset are shown in Figure 5, and the complete dataset is available online at [26].

### 4.2. Vision Transformers Implemented

Popular vision transformers include ViT-B and ViT-L architectures. Both architectures differ from each other with respect to the dimension of flattened patches *D* such that *D* equals 768 for ViT-B and 1024 for ViT-L. In this study, pre-trained ViT-B models are adopted as their lower dimension makes them faster to train. We employ two variants of the ViT-B model corresponding to the input patch size, including ViT-B/16 and ViT-B/32. The former refers to the input patch size of 16 × 16, whereas the latter corresponds to a 32 × 32 patch size. Smaller patch sizes are resource-intensive. The models are pre-trained on the 21k-ImageNet dataset, which consists of 21k classes and 14 million images. Furthermore, the architecture is fine-tuned on the ILSVRC-2012 ImageNet dataset consisting of 1k classes and 1.3 million images.

### 4.3. Experimental Design

The training and evaluation process for the CNN architectures comprised four stages:Stage 1: Pre-trained the state-of-the-art CNN architectures on the ImageNet dataset.Stage 2: Re-trained the architectures on our proposed road weather dataset using cross entropy loss function.Stage 3: Optimise the architectures using class weighted loss function.Stage 4: Optimise the architectures using focal loss function.Stage 5: Pre-trained the state-of-the-art Transformer vision models on the ImageNet dataset.Stage 6: Re-trained the architectures on our proposed multi-label road weather dataset.

In the first stage, the ImageNet [27] dataset is utilised to pre-train the CNN architectures: VGG19, GoogLeNet, ResNet-152, Inception-v3 and EfficientNet-B7. We chose these architectures due to their remarkable image classification performance on the ImageNet dataset [28,29]. The images are first resized into the required image size for the CNN architectures, e.g., 224 × 224 for most of the models except EfficientNet-B7 and Inception-v3, which require an input size of 600 × 600 and 299 × 299, respectively. Later, the models are pre-trained by setting the ‘pre-trained’ parameter in the models to True (in Pytorch).

In stage 2, the pre-trained models are re-trained on our proposed road weather dataset by replacing the number of outputs in the final fully connected layer of the CNN models with the number of weather classes. It is important to note that ‘icy’ road weather images were eliminated from our experiments as they were only three instances. Therefore, seven weather classes were utilised in our experiments to re-train the models. Only the last layers of the CNN architectures are optimised during the training process using cross entropy loss.

In the third stage, we update the cross entropy loss to incorporate the number of images in each class (i.e., class weighted loss function). This is important to reduce the bias of the majority classes of imbalanced datasets by providing higher weights to images from minority classes and lower weights to images from majority classes.

In the fourth stage, focal loss function is implemented to pay more attention to classes that are harder to learn, e.g., extreme (rare) weather conditions.

Convolution neural networks assign similar weight to all the pixels during classification, which might lead to inefficient results, especially in a complex road image with a lot of background noise. To tackle this, in the fifth stage, an attention mechanism is implemented using Vision Transformers (ViT), which are pre-trained on the ImageNet dataset. In the last stage, ViT models, namely ViT-B/16 and ViT-B/32, are re-trained on the proposed road dataset for multi-label weather detection.

### 4.4. Evaluation Protocol

The CNN architectures were trained and evaluated using five-fold cross-validation using Pytorch programming language. The optimal learning rate for the models was set to 0.001 and momentum was 0.9. A batch size of 32 and 50 epochs was utilised in training the models. It is worth mentioning that the results obtained by the training and validation set at each fold were averaged to evaluate the final performance of the models. We used the following evaluation metrics to compare the performance of models: accuracy and F1-score. Since the data are highly imbalanced, the F1-score is a better metric to evaluate the models. Vision Transformer models were trained and evaluated using exactly the same hyperparameter settings and the patch size of 16 × 16 and 32 × 32 for Vit-B/16 and Vit-B/32, respectively.

## 5. Results and Discussion

### 5.1. State-of-the-Art CNN Models

Table 3 shows the multi-label classification results for the pre-trained models using binary cross entropy loss. It can be seen that ResNet-152 outperforms the other state-of-the-art CNN models in both accuracy and F1-score using our multi-label road weather dataset, followed by VGG19 and EfficientNet-B7. ResNet-152 achieves an average validation accuracy of 87.73% and F1-score of 64.22%. This result is similar to previous studies [6,7,8,11], where ResNet-152 showed better performance compared to other CNN architectures. However, the F1-score is low due to the imbalanced nature of the dataset.

After optimising the models using the class weighted loss function to reduce bias produced by the majority classes, we observe the classification results in Table 4. The table shows the multi-label classification results for the pre-trained models with a class weighted loss function. Weights assigned to each class correspond to the *total number images* divided by *total images in that class*. It can be seen that by optimising the models with the class weighted loss function, performance has improved significantly. The best-performing model, ResNet-152, now has an average F1-score of 71%. The CNN models VGG19, GoogleNet, ResNet-152, Inception-v3, EfficientNet-B7 have increased by 0.29%, 2.37%, 1.4%, 1.72% and 1.71% for the weighted loss function as compared to binary cross entropy loss. The model GoogleNet seemed to have the most significant improvement of 2.37% in the training accuracy when introducing weighted loss function. However, the models VGG19 and EfficientNet-B7 are still the second and third-best models. The validation accuracy for all the models has also increased in Table 4, with GoogleNet attaining the most significant difference of 2.11%, while ResNet-152 is still the best-performing model with a validation accuracy of 88.48% for the weighted loss function.

When we focus on images that are difficult to classify, a focal loss function is used to optimise the models. Table 5 shows that by using the focal loss function, performance improves further. ResNet-152 still outperforms the other models with a 74.4% F1-score. However, the best overall improvement can be seen for the model GoogleNet with a 17.74% from binary cross entropy loss to focal loss function and a 4.45% increase from class weighted loss function to focal loss function. GoogleNet and Inception-v3 are now the second and third-best-performing models instead of VGG19 and EfficientNet-B7. The other models, ResNet-152, Inception-v3, EfficientNet-B7, have improved their validation accuracy for the focal loss function by 0.44%, 0.13%, 0.39% and 0.48% as compared to the weighted loss function. The focal loss function produces an overall better performing and generalised model since the average F score is significantly improved for all the models when introducing the focal loss function. The F score for focal loss has a significant improvement of 2.07%, 4.45%, 3.4%, 3.77% and 2.77% for VGG19, GoogleNet, ResNet-152, Inception-v3 and EfficientNet-B7, respectively, as compared to the class weighted loss function. The best-performing CNN model for all the loss functions is ResNet-152 with an F score of 64.22%, 71%, 74.4% for binary cross entropy, weight loss and focal loss function. However, the focal loss function seems to outperform all other loss functions by focusing more on hard instances. The class weighted loss function seems to be the second-best loss function as it assigns more weight to classes with fewer instances and significantly improves the F score for all the CNN models, respectively.

### 5.2. Vision Transformers

Lastly, given the highly imbalanced nature of our dataset, the results achieved so far are satisfactory. However, overcoming the limitations of the CNN model, the transformer vision model further incorporates attention mechanisms to the instances forcing the model to focus on relevant pixels only. Table 3, Table 4 and Table 5 show the results obtained from the pre-trained ViT models—ViT-B/16 and ViT-B/32—for binary cross entropy loss, weighted loss and focal loss function, respectively. It can be seen that incorporating attention mechanisms in the architecture has significantly improved the overall accuracy as well as F-score for our multi-label road dataset. In Table 3, the validation accuracy achieved for ViT-B/16 is 91.92% along with an 81.22% F-score, and the validation accuracy achieved for ViT-B/32 is 91.45% along with 80.48%, both of which outperform all the CNN models. Nevertheless, ViT-B/16 slightly outperforms ViT-B/32 and significantly outperforms the best-performing focal-loss-based ResNet-152 with a 3.72% increase in the validation accuracy and 6.82% increase in the F-score for our given dataset. In Table 4, vision transformers still outperform the best-performing weighted loss function-based ResNet-152 significantly, with an F score of 79.18% and 77.912% for ViT-B/16 and ViT-B/32. Similarly, in Table 5 for focal loss function, ViT-B/16 and ViT-B/32 achieve an average F score of 80.23% and 80.25% as compared to 74.4% for ResNet-152. Therefore, it is quite evident that Vision Transformers outperform the CNN models for our given dataset significantly, with ViT-B/16 being the best-performing model.

## 6. Conclusions

Intelligent weather detection is important to support safe driving and effective management of the transport network. Previous computer vision studies perform multi-class weather classification, which is not always appropriate and reliable for road safety, as multiple weather conditions are likely to occur simultaneously. In addition, the majority of them use balanced randomly selected outdoor images, which are unrepresentative of the real-world frequency of weather types and the unbalanced nature of road weather data. In this paper, we have introduced multi-label deep learning architectures for road weather classification, i.e., VGG19, GoogleNet, ResNet-152, Inception-v3, and EfficientNet-B7. To adequately evaluate their performance, we have created a multi-label road weather dataset using naturalistic road clips captured by onboard cameras. The dataset consists of road images captured at different road types, different lighting conditions and different weather and road surface conditions. Due to the imbalanced nature of the dataset, we improved model performance using class weighted and focal loss functions to handle rare weather conditions and hard-to-classify images. Results show significant classification improvement when higher weights are assigned to rare weather conditions (class weighted loss function), e.g., snowy and icy weather, thereby reducing overfitting on frequently occurring weather conditions, such as sunny and cloudy. Additionally, further improvement is observed when the models are forced to focus more on hard-to-classify weather images (focal loss function). Furthermore, we explore attention mechanisms for pixel-based dynamic weight adjustment and segmentation to improve the models’ performance. This is essential in separating the road layouts from the background and providing higher weights to pixels depending on the weather conditions. For example, cloudy weather can be easily recognised by analysing the background (clouds) and wet weather by analysing the road. This was achieved using transformer vision models, ViT-B/16 and ViT-B/32, which outperformed all other CNN architectures. For future work, Grad-CAM interpretation can be implemented to observe an in-depth visual explanation to understand the learning process of ViT models under these scenarios.

## Figures and Tables

**Figure 1 sensors-23-00798-f001:**
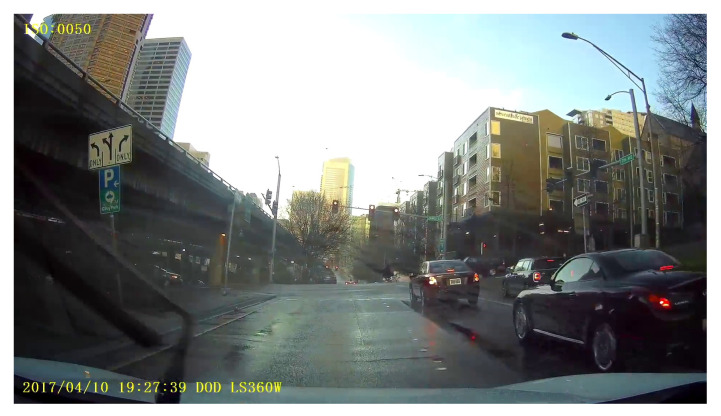
Multiple weather conditions (sunny and wet) existing in a single image.

**Figure 2 sensors-23-00798-f002:**
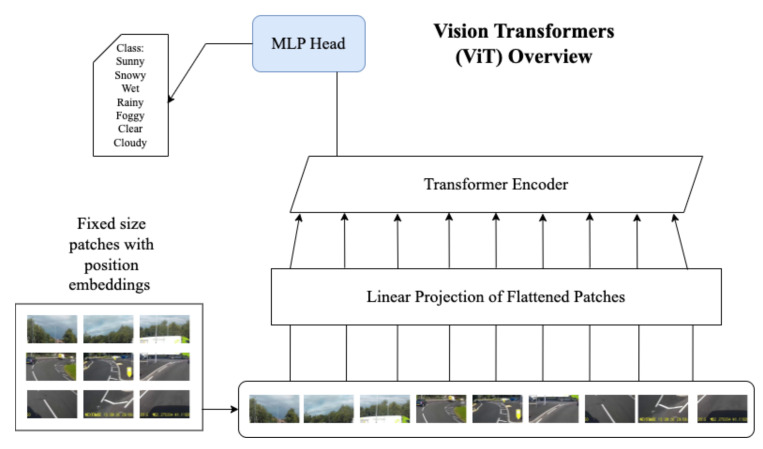
Transformer vision model architecture overview.

**Figure 3 sensors-23-00798-f003:**
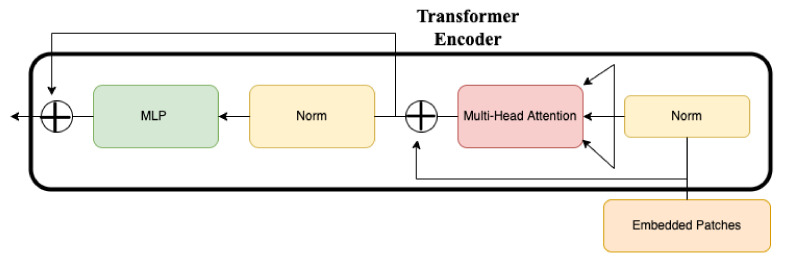
Overview of the transformer encoder.

**Figure 4 sensors-23-00798-f004:**
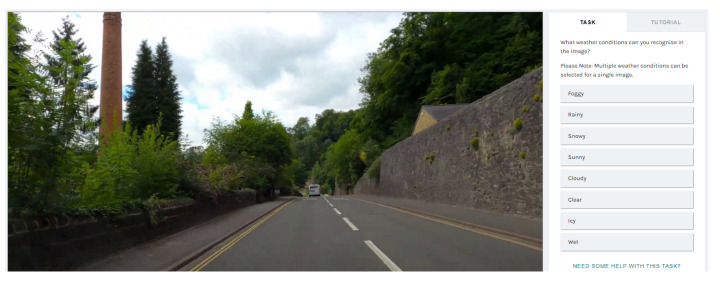
A screenshot of using Zooniverse to annotate road weather images.

**Figure 5 sensors-23-00798-f005:**
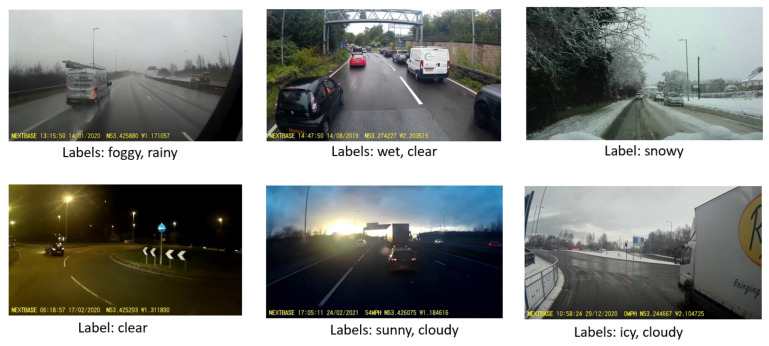
Six samples of weather images from our multi-label road weather dataset.

**Table 1 sensors-23-00798-t001:** State-of-the-art CNN models assessed in this study.

Model	Author	Year	Number of Layers	Input Image Size
VGG19	Oxford University Researchers [18]	2014	19 layers	224 × 224
GoogleNet	Researchers at Google [19]	2015	22 layers	224 × 224
ResNet-152	He et al. [20]	2015	152 layers	224 × 224
Inception-v3	Szegedy et al. [21]	2016	48 layers	299 × 299
EfficientNet-B7	Tan et al. [22]	2019	813 layers	600 × 600

**Table 2 sensors-23-00798-t002:** Class distribution of the proposed road weather dataset.

Class	Number of Instances
Clear	1299
Sunny	1184
Cloudy	626
Wet	369
Snowy	147
Rainy	84
Foggy	78
Icy	3

**Table 3 sensors-23-00798-t003:** Multi-label classification results for road weather detection using simple binary cross entropy loss function (best performance in bold).

Model	Avg Training Accuracy	Training SD	Avg Validation Accuracy	Validation SD	Avg F Score	F-Score SD
VGG19	84.19	0.005	85.14	0.002	58.50	0.008
GoogleNet	84.42	0.009	85.08	0.006	50.52	0.012
ResNet-152	87.58	0.003	87.73	0.005	64.22	0.014
Inception-v3	84.23	0.008	84.80	0.006	50.56	0.004
EfficientNet-B7	85.11	0.003	86.03	0.003	56.09	0.007
ViT-B/16	93.52	0.0118	**91.92**	0.0088	**81.22**	0.0182
ViT-B/32	**94.65**	0.0262	91.45	0.0065	80.48	0.0115

**Table 4 sensors-23-00798-t004:** Multi-label classification results for road weather detection using class weighted loss function to force models to handle rare weather conditions (best performance in bold).

Model	Avg Training Accuracy	Training SD	Avg Validation Accuracy	Validation SD	Avg F Score	F-Score SD
VGG19	84.48	0.002	85.35	0.005	64.21	0.015
GoogleNet	86.79	0.002	87.19	0.003	63.54	0.010
ResNet-152	88.98	0.001	88.84	0.003	71.00	0.011
Inception-v3	85.95	0.004	86.87	0.004	62.52	0.009
EfficientNet-B7	86.82	0.002	87.24	0.005	63.38	0.007
ViT-B/16	95.97	0.3579	**90.95**	0.0076	**79.18**	0.0211
ViT-B/32	**98.66**	0.0178	90.48	0.0043	77.912	0.0073

**Table 5 sensors-23-00798-t005:** Multi-label classification results for road weather detection using focal loss function to force models to handle difficult-to-classify weather images (best performance in bold).

Model	Avg Training Accuracy	Training SD	Avg Validation Accuracy	Validation SD	Avg F Score	F-Score SD
VGG19	83.90	0.003	84.85	0.005	66.28	0.012
GoogleNet	87.22	0.002	87.63	0.004	67.99	0.014
ResNet-152	89.44	0.004	88.71	0.007	74.40	0.010
Inception-v3	85.91	0.002	87.26	0.002	66.29	0.006
EfficientNet-B7	87.48	0.001	87.72	0.005	66.15	0.008
ViT-B/16	93.95	0.02942	**91.26**	0.0059	80.23	0.0077
ViT-B/32	**94.80**	0.3387	91.23	0.0050	**80.25**	0.0125

## Data Availability

Publicly available datasets were analyzed in this study. This data can be found here: [https://drive.google.com/file/d/1e7NRaIVX6GNqHGC_aAqaib_DU_0eMVRz, accessed on 5 December 2022].

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
