# Peer review of "Deep Learning with Attention Mechanisms for Road Weather Detection"

_sensors, 2023, doi:10.3390/s23020798_

Round 1

Reviewer 1 Report

1. Your manuscript contains sections of bulk citation.For example:

Deep learning has emerged as one of the main approaches used for automatic weather recognition[3-6].

It is important that each article is discussed in detail to ensure that the relevance of the source is clear to readers, and to provide suitable support to your manuscript.Please re-assess all highly relevant to your manuscript and that they are discussed in detail.

2. I don't see the category of icy in Table 2, but I see an example of icy in Figure 5. Please carefully proofread the dataset description.

3. In the part of optimizing the architecture, it is recommended to increase the comparative demonstration of ablation experimental data.

4.The first paragraph may present a much broad and comprehensive view of the problems related to your topic with citations to authority references: Novel visual crack width measurement based on backbone double-scale features for improved detection automation, Engineering Structures 2023.

Reviewer 2 Report

In the paper “Looking Deeper into Images for Autonomous Road Weather Detection” the authors propose a comparison between ViT classifier and state-of-the-art CNN counterparts to automatically detect road weather. 

The methodological choices are well explained in the paper. 

Below are some comments: 

The related work section is quite short. The authors should at least mention iGPT and highlight the differences with ViT, moreover a mention of the ViT variants should be added. 

The authors could expand the conclusion by discussing if there are advantages of combining the transformer with convolution for the task described. 

Figure 3 should be improved, a road photo with less white background should be chosen so that all the patches are clearly visible, the space between patches should have the same offset 
